# Signatures of correlation of spacetime fluctuations in laser interferometers

**B. Sharmila** [1] ✉, **Sander M. Vermeulen** [2] **& Animesh Datta** [1]

Spacetime fluctuations (SFs), a common feature of proposed gravity models, could be detected using laser interferometers. To advance this effort, we provide the correspondence between expected interferometer output signals and gravity models. We consider three classes of SFs, characterised by the decay behaviours and symmetries of their two-point correlation functions. For each, we identify the low- and high-frequency behaviour of the outputs and their dependence on the interferometer's length. Capturing these requires sensitivity over a broad frequency range that spans the light-round-trip frequency, as provided by laboratory-scale setups, whereas detecting the presence or absence of SFs can be done with high narrowband sensitivity at the light-round-trip frequency. Our approach applies to interferometers with arm cavities, such as the km-long LIGO detectors, and those without, like laboratory-scale setups QUEST and GQuEST. Finally, we constrain the strength and correlation scale of SFs by comparing our modelled signals with experimental data.

A wide spectrum of ideas has been considered[1,2] to understand the fundamental nature of gravity. Some, such as the idea of spacetime fluctuations (SFs), form a *leitmotif* in this effort. Since its first proposal by Wheeler[3], SFs have been extensively examined[4] in the context of a quantum description of gravity, as well as in different semiclassical models of gravity[5–11] and the study of stochastic gravitational waves[12,13]. Spacetime has thus been hypothesised to be, for instance, classical but stochastic[14], or classical but emerging from underlying quantum entanglement[15], or holographic and having quantum perturbations[16]. These hypotheses suggest different mathematical forms for the correlation functions of SFs.

The scale of the correlations in SFs differs widely across models, ranging from the Planck length scale in effective field theories[17] to long-range correlations in holographic models[18,19]. The latter hypothesise observable effects such as the violation of Lorentz invariance[20], gravitational decoherence[2,21], blurring of astronomical objects[22], and interferometric noise[23–25]. Of these, the last has garnered attention through the development of gravitational-wave detectors[26], and the Holometer[27,28] experiment. Recently, sophisticated laboratory-scale Michelson laser interferometers (MLIs), such as QUEST[29] and GQuEST[25], which incorporate new quantum technologies, aim to search for SFs.

Unambiguous detection of SFs would constitute a breakthrough in understanding the fundamental nature of gravity. To enable such a detection, experimental designs require estimates of the strength and bandwidth of the expected interferometric output signal so they can be suitably optimised. Further, computing the interferometric output signal can help in understanding how SF signals, which are typically broadband, can be distinguished from instrumental noise. Moreover, quantitative theoretical predictions of the output signal are essential either to rule out the presence of SFs in interferometric data or, if the underlying gravity theory includes free parameters, to constrain those parameters. While many theoretical works have modelled the interferometric output signal to identify possible SF signatures[10,16,17,30–32], they only offer model-specific predictions.

We take a more expansive approach in studying SFs. Irrespective of the classical or quantum description of a phenomenon, correlations in physical processes tend to decay either exponentially or polynomially with increasing separation $r$ between two spacetime points[33,34]. Exponential decay of correlations typically emanates from underlying physics that is short-ranged. In the context of gravity, this captures both quantum[15] and semiclassical[11] models. On the other hand, polynomial decay of correlations of the form $r^{-\eta}$ ($\eta \in \mathbb{R}^+$),

[1]Department of Physics, University of Warwick, Coventry, UK. [2]Division of Physics, Mathematics and Astronomy, California Institute of Technology, Pasadena, USA. ✉e-mail: Sharmila.Balamurugan@warwick.ac.uk

emanate from long-range interactions. In the context of gravity, such correlations of SFs are expected to decay as a reciprocal of distance, i.e., $1/r$[16,17,22,35] or possibly with a higher inverse power-law such as $r^{-4/3}$[36]. Furthermore, internal symmetries of the spacetime metric are reflected in induced symmetries of the correlation function, such as factorisation into spatial and temporal parts. This factorisation occurs in SF models such as the Oppenheim model[10], a generalised Károlyházy model[36], the continuous spontaneous localisation model[37,38], and the Diósi-Penrose model[39–41].

Thus motivated, we consider three possible classes of two-point correlation functions of the SFs: (a) factorised into spatial and temporal correlations, both decreasing with increasing spatial and temporal separation (for instance, a semiclassical model[10]); (b) an inverse of the separation between the two spacetime points (for instance, an SF model obeying the wave-equation in $3+1$ dimensions[16,22,35]), with subclasses refining the definition of the separation as either a spatial or a spacetime separation; and (c) an exponential decay with the separation between the two spacetime points (for instance, due to quantum entanglement[15], and in semiclassical[11] models), with subclasses as in (b). Finding experimental evidence for or against one of the above classes of correlation functions describing SFs would constitute a breakthrough in elucidating the fundamental nature of gravity.

In this work, corresponding to each class of SFs, we identify three characteristic signatures: the low-frequency behaviour, the high-frequency behaviour and the $\mathcal{L}$ dependence of the interferometric output signal, where $\mathcal{L}$ is the arm length of the MLI. In order to observe all three signatures, it is sufficient if the MLIs are sensitive over two decades spanning the light round-trip frequency $f_{LRT} = c/(2\mathcal{L})$. However, even MLIs that are sensitive in a shorter span that includes $f_{LRT}$, could allow observation of the three signatures to a limited extent. The projected sensitivities of laboratory-scale MLIs such as QUEST and GQuEST span the $f_{LRT}$, unlike LIGO. Therefore, we find that these laboratory-scale MLIs allow observation of more signatures than LIGO in principle, thus providing more information on the class of the underlying correlation function and aiding in distinguishing between the classes (a)–(c).

We also show that MLIs with arm cavities have a significant advantage in detecting the bare presence or absence of SFs. This is due to the peak in the interferometric output signal for MLIs with arm cavities at their $f_{LRT}$. We note that for LIGO, $f_{LRT} \approx 37.5$ kHz, which is outside the frequency span of the publicly available data. GWTC-3 run[42] in LIGO reports data only up to 5 kHz. We await new results from experimental runs that report at a higher frequency range.

Our approach is agnostic to the microscopic origins of the SFs. It is important to emphasise that the microscopic origins of the SFs differ widely between the different gravity models, ranging from quantum entanglement of degrees of freedom resulting in suitable moments of the metric[15] to *geontropic*[16] or stochastic[11] fluctuations of the metric itself. Our approach requires as input only the two-point correlation of the spacetime metric from any model of gravity and the geometry of the MLI. This enables us to compute the output signal power spectral density (PSD) of the MLI on which all our conclusions are based. Our approach also accommodates computing the signal PSD for MLIs with arm cavities, such as LIGO. Whether the presence of arm cavities provides any advantage towards detection has been much debated lately[16,18,43]. This debate even led to the exclusion of arm cavities from experimental designs of recent MLIs[25,28,29]. We now settle the debate for any SF from classes (a)–(c).

In addition, we compare the computed output signal PSD with the measured output signal from the Holometer and QUEST MLI experiments to constrain the free parameters in classes (a)–(c). We also compare our constraints with prior ones[16].

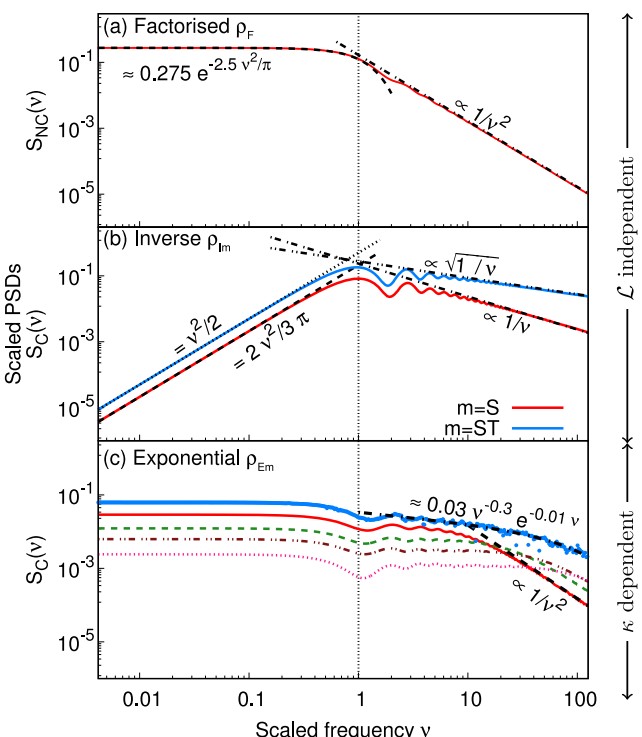

**Fig. 1 | MLI output for three correlation classes.** Output signal power spectral densities (PSDs) of an MLI due to SFs. For correlation class (**a**), *factorised* $\rho_F$, the scaled PSD $S_{NC}(\nu) = \frac{cS(f)}{\Gamma_S \mathcal{L}^3}$ vs. $\nu = \frac{\pi f}{2 f_{LRT}}$ is plotted. For classes (**b**), *inverse* $\rho_{Im}$, and (**c**) *exponential* $\rho_{Em}$ with $m =$ S, ST, the scaled PSD $S_C(\nu) = \frac{cS(f)}{\Gamma_S \ell_r \mathcal{L}^2}$ vs $\nu$ is plotted. Here, $m =$ S ($m =$ ST) denotes correlations depending on spatial (spacetime) separation. In (**c**), PSDs corresponding to $\rho_{ES}$ for $\kappa = \ell_r/\mathcal{L} = 0.025$ (red solid), $\kappa = 0.01$ (green dashed), $\kappa = 0.005$ (brown dot-dashed), and $\kappa = 0.0025$ (pink dotted), demonstrate dependence on $\kappa$. The PSD corresponding to $\rho_{EST}$ for $\kappa = 0.025$ (blue points) is also plotted in (**c**). Small and large $\nu$ trends are as indicated by black dashed/dotted lines in (**a**)–(**c**). The black vertical line marks $\nu = 1$.

## Results

In this paper, we present three main results, namely, distinguishing between correlation classes (a)–(c) using three characteristic signatures identified in the output signal PSD of the MLI without arm cavities, the advantage enjoyed by the MLI with arm cavities in detecting the bare presence or absence of spacetime fluctuations, and constraining the free parameters in the correlation functions from different gravity models using the measured output signal from the Holometer and QUEST MLI experiments. To this end, we use the approach described in the "Methods" section to obtain the PSD of the optical path difference between the two arms of the MLI in the presence of random fluctuations $w(\mathbf{r})$ at $\mathbf{r} \equiv (t, x, y, z)$ in the spacetime metric (Eq. (4)). Thus obtained, the interferometric output signal PSD for MLI without (Eq. (15)) and with (Eq. (22)) arm cavities involves the correlation function $\overline{w(t_1, \vec{r}_1) w(t_2, \vec{r}_2)} = \Gamma_S \rho(ct_{12}, \vec{r}_{12})$ (Eq. (8)) where $X_{12} = X_1 - X_2$ ($X = t, \vec{r}$) with $\vec{r}_i \equiv \{x_i, y_i, z_i\}$ ($i =$ 1, 2) and $\Gamma_S$ is the strength of the SFs. Subsection "Correlation classes" in the "Methods" section lists the correlation functions defining the three correlation classes (a)–(c). Further, for the ease of comparison, we consider dimensionless frequency and PSDs defined in the "MLI output signal PSDs" subsection : $\nu$ (Eq. (24)), $S_{NC}$ (Eq. (25)), and $S_C$ (Eq. (26)). Note that we have two types of dimensionless PSDs: $S_{NC}$ independent of the correlation length $\ell_r$, catering to correlation class (a) which is also independent of $\ell_r$, and $S_C$ which depends on $\ell_r$ for correlation classes (b) and (c). From Eqs. (15, 17 and 22), it is clear that both $S_{NC}(\nu)$ and $S_C(\nu)$ are independent of $\Gamma_S$.

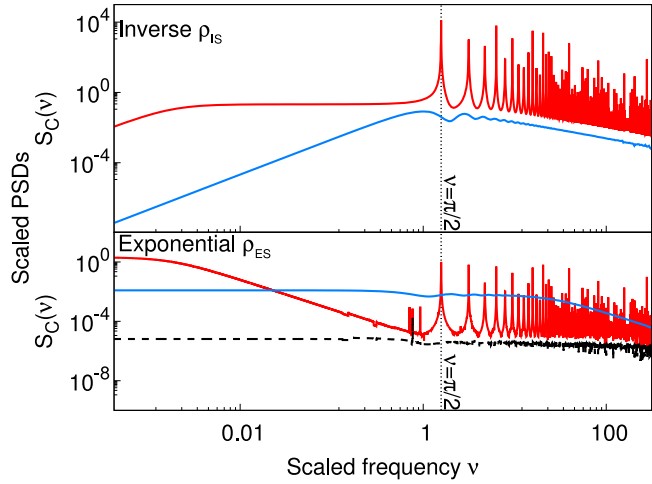

**Fig. 2 | Output of MLI with arm cavities.** Scaled PSD $S_C(\nu)$ corresponding to LIGO (red) with $\mathcal{L} = 4000$ m, QUEST (blue) with $\mathcal{L} = 3$ m and a hypothetical MLI without arm cavities (black dashed) with $\mathcal{L} = 4000$ m vs scaled frequency $\nu$ is plotted for (top) *inverse* $\rho_{IS}$, and (bottom) *exponential* $\rho_{ES}$ correlation functions with $\ell_r = 0.03$ m.

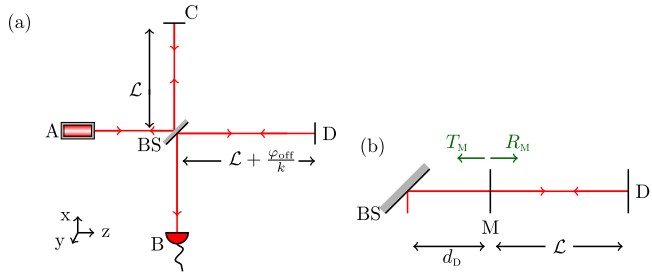

**Fig. 3 | Geometry of the MLI setup. a** A schematic diagram of the Michelson laser interferometer (MLI). Each of the two interferometers in the Holometer[28], QUEST[29], and GQuEST[25] is an MLI. **b** Fabry-Pérot arm cavity design used in LIGO-type interferometers. Mirror M is introduced in each of the two arms of the MLI to create optical resonators (illustrated for arm D) with power transmissivity $T_M = 1 - R_M$.

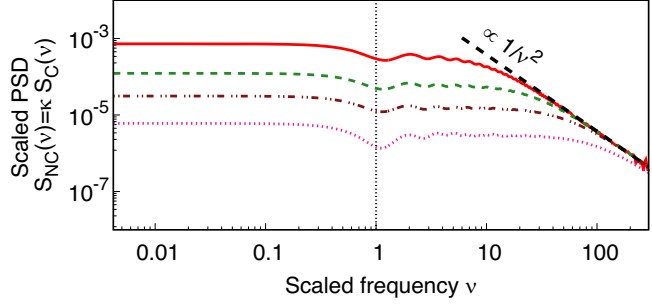

**Fig. 4 | $\kappa$-dependence of MLI output in class (c).** Scaled output signal PSDs $S_{NC}(\nu) = \kappa\, S_C(\nu)$ vs $\nu$ are plotted for *exponential* $\rho_{ES}$ corresponding to $\kappa = 0.025$ (red solid), $\kappa = 0.01$ (green dashed), $\kappa = 0.005$ (brown dot-dot-dashed), and $\kappa = 0.0025$ (pink dotted), demonstrating the $1/\nu^2$ behaviour at the high-frequency limit.

We present the output signal PSDs for an MLI without (see Fig. 1(a)–(c)) and with arm cavities (see Fig. 2) for $\rho$ corresponding to correlation classes: (a) factorised $\rho_F$ (Eq. (10)), (b) inverse $\rho_{Im}$ (Eqs. (11, 12)), and (c) exponential $\rho_{Em}$ (Eqs. (13, 14)) where $m = S$, ST, with $m = S$ ($m = ST$) denoting dependence on spatial (spacetime) separation.

## Distinguishing correlation functions

To list the three characteristic signatures of the interferometric output signal PSD corresponding to each correlation class, we first consider the case of a simple MLI without arm cavities, as in Fig. 3(a). We present the analytical expressions for the PSDs corresponding to each class (For more details, see Sec. V of the Supplementary Information for class (a) and Sec. VI of the Supplementary Information for classes (b) and (c)).

*Low-frequency limit $\nu \ll 1$:* For class (b) (Fig. 1 (b)), the PSDs are proportional to $\nu^2$ with $S_C(\nu = 0) = 0$ in this limit. We analytically find the constants of proportionality as $2/(3\pi)$ and $1/2$ for $\rho_{IS}$ and $\rho_{IST}$ respectively (see, Sec. VI of the Supplementary Information for details). This can also be understood intuitively, especially in the case of $\rho_{IS}$. In this particular case, the fluctuations satisfy the wave equation and therefore, the 4-d Fourier transform (Eq. (18)) of $\rho_{IS}$ involves a Dirac delta function that forces $2\pi f = c|\vec{k}_1|$ (see Eq. (81) of the Supplementary Information for the exact form). This leaves the interferometer response function (Eq. (19)) as the only frequency-dependent contribution to the PSD, which, up to a constant factor, is $\mathcal{L}^2 f^2 |T_+^{(x)} - T_+^{(z)}|^2/4$ in this limit. The $f^2$ in this term produces the quadratic trend of PSD in class (b).

For classes (a) and (c), the PSDs are almost flat in a log-log plot in the limit $\nu \ll 1$ in Fig. 1 (a), (c). We find the numerical fit in this limit for both cases to be of the form $e^{-\alpha \nu^2}$ where $\alpha \in \mathbb{R}$ is found numerically. In class (a), the exact numerical fit is $S_{NC}(\nu \ll 1) \approx 0.275 e^{-\frac{5\nu^2}{2\pi}}$. This fit is independent of the value of $\mathcal{L}$ (see Sec. V of the Supplementary Information for details). In class (c), the value $\alpha$ in $e^{-\alpha \nu^2}$ depends on the specific value of the ratio $\kappa = \ell_r/\mathcal{L}$. For instance, $\alpha = 4/\pi$ for $\kappa = 0.01$ (see Sec. VI of the Supplementary Information for details).

*High-frequency limit $\nu \gg 1$:* As is evident from Fig. 1 (a)–(c), the PSDs for all correlation classes decrease with increasing $\nu$. However, the rates at which they decay are starkly different. In Sec. VI of the Supplementary Information, we analytically justify why we find the following decay rates of $S_C(\nu)$: $\propto 1/\nu$ for $\rho_{IS}$, $\propto 1/\sqrt{\nu}$ for $\rho_{IST}$, and $\propto 1/\nu^2$ for $\rho_{ES}$. We also present $S_{NC}(\nu)$ vs $\nu$ for $\rho_{ES}$ in Fig. 4. Though the PSD in this case is evidently dependent on $\kappa$, and therefore $\ell_r$, we use $S_{NC}(\nu)$ to showcase the high-frequency behaviour, which is always $\propto 1/\nu^2$ irrespective of the value of $\kappa$ (as this is not very apparent from Fig. 1(c)). We also note here that the onset of this decay is delayed as the ratio $\kappa$ decreases.

The other decay rates, such as $S_{NC} \propto 1/\nu^2$ for class (a) and $S_C(\nu) \approx 0.03\, \nu^{-0.3}\, e^{-0.01\nu}$ for $\rho_{EST}$, are obtained through numerical fits. Of these, the functional form of the latter is obtained by considering the possible form of the cosine transform of $\rho_{EST}$ (see Sec. VI of the Supplementary Information for details).

*Dependence on $\mathcal{L}$ and $\kappa$:* For class (a), we analytically show that $S_{NC}(\nu)$ is independent of $\mathcal{L}$ in Eq. (68) of the Supplementary Information, and that $S_C(\nu)$ for class (b) is independent of $\mathcal{L}$ in Eqs. (69, 70) of the Supplementary Information. For class (c), we analytically show that $S_C(\nu)$ depends on $\kappa$ in Eqs. (74-76) of the Supplementary Information. Figure 1(c) illustrates this $\kappa$-dependence of the PSDs.

As shown above, the different classes of spacetime fluctuations produce three characteristic signatures in their corresponding output signal PSDs. These can be used to identify the nature of the underlying SFs from interferometric data. Here, it is worth pointing out another feature that is present in the scaled PSDs of the three classes is the oscillatory behaviour at $\nu > 1$. However, this feature originates from the interferometer response (See Eqs. (49-50) of the Supplementary Information for details) of the different classes and is not a characteristic signature of the different classes.

To observe all three signatures characteristic of each correlation class, the interferometer should ideally be sensitive in the range of $\nu \approx 0.1$ to $\nu \approx 10$. In QUEST with $\mathcal{L} = 3$ m (resp., GQuEST with $\mathcal{L} = 5$ m), the sensitive bandwidth is designed to span from 1 MHz to 250 MHz (resp., 8 MHz to 40 MHz)[25], with a corresponding span of $0.03 \leqslant \nu \leqslant 78$

(resp., $0.42 \leqslant \nu \leqslant 2.1$). This illustrates that the bandwidths of both QUEST and GQuEST would allow observation of all three signatures, although the narrower bandwidth of GQuEST could limit the observation of the low- and high-frequency signatures to some extent.

On the other hand, experimental data from LIGO covers the frequency range only from about 10 Hz to 10 kHz, corresponding roughly to $0.0004 < \nu < 0.4$. However, for completeness, we discuss the low- and high-frequency behaviour of the PSD for LIGO in Sec. VIII of the Supplementary Information.

To summarise, QUEST and GQuEST, with their broader bandwidths, allow observation of all the characteristic signatures that could help in distinguishing between correlation functions using their interferometric PSD data.

### Detecting SFs

To highlight the advantage LIGO enjoys in detecting the SFs, we list three key features of the interferometric output signal of the MLIs with arm cavities (see Fig. 3(b) for the geometry, and see Fig. 2 for the PSDs) for the different correlation classes. For further details, see Sec. VIII of the Supplementary Information.

(1) For any fluctuation described by a Gaussian random process, the signal PSD (Eq. (22)) of MLIs with arm cavities has peaks at $\nu = m\pi/2$, for $m = 1, 2, 3, \cdots$, of magnitude $T_M^4/(1 - \sqrt{R_M})^6$. For LIGO with $T_M = 1 - R_M = 0.014$ of the input mirror of the arm cavity, the magnitude of the peak is $\approx 3.2 \times 10^5$.

(2) For class (b), $S_C$ for an MLI without arm cavities is independent of $\mathcal{L}$, with a global maximum at $\nu = 1$. Therefore, in this class, the Fabry-Pérot cavity response (Eq. (23)) enhances the signal strength (Eq. (22)) at every frequency $\nu$. The strongest signal is at $\nu = \pi/2$ (equivalently, $f \approx 37.5$ kHz for LIGO), as expected. This is illustrated in the top panel of Fig. 2. This is also consistent with prior work[16]. A similar argument holds for class (a) as its $S_{NC}$ for an MLI without arm cavities is also independent of $\mathcal{L}$.

(3) For class (c), $S_C$ is directly proportional to $\kappa$. We compare $S_C$ for a given $\ell_r$ in two different setups, with and without arm cavities and with different arm lengths: LIGO with $\mathcal{L} = 4$ km, and QUEST with $\mathcal{L} = 3$ m. It is evident from the values of $\mathcal{L}$ that the ratio $\kappa$ in QUEST is far greater than that in LIGO for a given $\ell_r$. Thus, $S_C \propto \kappa$ implies that the LIGO signal is reduced with respect to that of QUEST by the factor $\approx 10^{-3}$ (i.e., the ratio $\mathcal{L}$ of QUEST to that of LIGO). This should be considered in conjunction with Feature 1 in this list (the presence of a peak at $\nu = \pi/2$ in the LIGO signal is always enhanced by the Fabry-Pérot cavity gain $\approx 10^5$). Therefore, when measuring SF with any $\ell_r$ in these setups, $S_C$ at $\nu = \pi/2$ of LIGO has a peak that exceeds the $S_C$ of QUEST. This is illustrated in the bottom panel of Fig. 2.

It is thus evident that for the classes (a)−(c) considered, LIGO has a clear advantage over QUEST and GQuEST in detecting the presence of SFs.

It is also evident from our arguments that this advantage of LIGO is not guaranteed for all correlation functions. For instance, let us consider a class of correlation functions for which $S_C \propto \kappa^2$ for an MLI without arm cavities. For such a class of correlation functions, using arguments similar to those used in discussing Feature 3, we can see that the peak in the LIGO signal does not exceed the PSD of QUEST. Therefore, LIGO does not enjoy an advantage in detection for such a class. This line of thinking might help in understanding some gravity models that predict LIGO should not enjoy any advantage in detecting SFs[18,43].

### Comparison with experimental data

We compare our results with experimental data from the Holometer [[44], Fig. 12], a now-retired MLI experiment with $\mathcal{L} = 40$ m, and QUEST [[45], Fig. 4], a table-top MLI experiment ($\mathcal{L} = 1.8$ m) that has just produced its first data. Note that we use QUEST data from the full measurement bandwidth, only part of which is published in Fig. 4 of[45]. While both the Holometer and QUEST experiments comprise twin MLIs designed to be cross-correlated, we restrict the present comparison to single-MLI PSDs for generality, avoiding additional assumptions on the correlation length of the SFs. We also do not consider data from LIGO or other gravitational-wave (GW) detectors, as these instruments operate and record data in a bandwidth that spans a relatively small range of $\nu$ and does not include the highest magnitude of the signal PSD, which is at $\nu = \pi/2$.

The data from the Holometer and QUEST is obtained using the measurement technique in ref. 46, so the output signal sensitivity is improved by removing quantum shot noise, and potentially revealing stationary signals and classical noise produced in the MLI. The data is dominated by classical noise, and no signal was detected (i.e., the three characteristic signatures mentioned earlier cannot be discerned in the data). Thus, we can constrain the two parameters $\Gamma_S$ and $\kappa$ (equivalently, $\ell_r$). To obtain the constraints, we consider the magnitude of the PSD $S(\nu = \nu_i)$ of the SFs described by each correlation class to be smaller than the experimentally measured spectral density $S_{meas}(\nu = \nu_i)$, at frequencies $\{\nu_i\}$. This experimentally measured $S_{meas}(\nu = \nu_i)$ contains negligible measurement uncertainty for the data considered. For instance, in the Holometer (respectively, QUEST) experiment, the constraint is obtained by $S(\nu = \nu_i) \leqslant S_{meas}(\nu = \nu_i)$ at every $\nu_i$ in the range $0.4 \leqslant \nu \leqslant 4$ (resp., $0.38 \leqslant \nu \leqslant 1.8$). The constraints are listed in Table 1 and plotted in Fig. 5 for classes (a)−(c). It is to be noted that the plot (Fig. 5) is a log-log plot of the parameter space $\{\Gamma_S, \kappa\}$. Each grid line on the $\Gamma_S$-axis marks an increase by a factor of 10, while that on the $\kappa$-axis corresponds to an increase by a factor of $10^{0.2} \approx 1.6$.

*(a) Factorised correlation function:* Considering $\rho_F$ (Eq. (10)) is independent of $\ell_r$, the constraint obtained involves only $\Gamma_S$. We know that $S_{NC}$ is independent of $\Gamma_S$ and $\mathcal{L}$ from Eq. (68) of the Supplementary Information. Therefore, the constraint is given by

$$\Gamma_S \leqslant \min \left\{ \frac{c\, S_{meas}(\nu_i)}{\mathcal{L}^3\, S_{NC}(\nu_i)} \right\}. \tag{1}$$

The constraints obtained using the above expression are quoted in Table 1 and plotted in Fig. 5 (a). We note that we consider $10^{-4} \leqslant \kappa \leqslant 5 \times 10^{-2}$. This range is chosen such that the two following conditions are satisfied: (1) $\kappa \ll 1$ to ensure that the path difference remains a stationary random process, and (2) $\kappa \gg \lambda/\mathcal{L}$, where $\lambda$ is the wavelength of light ($\sim 10^{-6}$ m for Holometer and QUEST), to ensure that the eikonal approximation remains valid. There are no constraints on the fluctuation strength considering $\rho_F$ from prior work.

*(b) Inverse correlation function:* The constraint in each subclass is given by

$$\Gamma_S \kappa \leqslant \min \left\{ \frac{c\, S_{meas}(\nu_i)}{\mathcal{L}^3\, S_C(\nu_i)} \right\}, \tag{2}$$

where $S_C(\nu)$ is given by Eqs. (69,70) of the Supplementary Information with $m = S$ (respectively, $m = ST$) for the subclass (b1) (resp., (b2)). Equation (2) yields the constraints listed in Table 1 and plotted in Fig. 5(b1), (b2).

In the case of the subclass (b1), the constraints are compared with the constraint $\alpha \leqslant 0.6$ in the Pixellon model[16]. This is because

$$\alpha \equiv \frac{2\Gamma_S \kappa \mathcal{L}}{3\sqrt{2\pi}\,\ell_P}, \tag{3}$$

where $\ell_P$ is the Planck length. In[16], the constraint on $\alpha$ is obtained by comparing the expected signal PSD with the cross-spectral density (CSD) data from the twin interferometers in the Holometer ($\sim 10^{-41}$ m$^2$ Hz$^{-1}$). This is in contrast to our comparison with the PSD data from a single interferometer ($\sim 10^{-39}$ m$^2$ Hz$^{-1}$ for the Holometer),

**Table 1 | Constraint on $\Gamma_S$ and $\kappa$ using experimental data from the Holometer and QUEST experiments for different correlation classes**

| Correlation class | | Experiment | | Prior bounds |
|---|---|---|---|---|
| | | Holometer | QUEST | |
| (a) Factorised $\rho_F$ | | $\Gamma_S \leqslant 1.628 \times 10^{-34}$ | $\Gamma_S \leqslant 5.154 \times 10^{-33}$ | None |
| (b) Inverse $\rho_{Im}$ | (b1) Spatial separation $m = $ S | $\Gamma_S\kappa \leqslant 6.075 \times 10^{-35}$ or $\alpha \equiv \frac{2\Gamma_S\kappa\mathcal{L}}{3\sqrt{2}\pi\ell_p} \leqslant 40$ | $\Gamma_S\kappa \leqslant 7.022 \times 10^{-33}$ or $\alpha \leqslant 208$ | $\alpha \leqslant 0.6$[16] |
| | (b2) Spacetime separation $m = $ ST | $\Gamma_S\kappa \leqslant 2.264 \times 10^{-35}$ | $\Gamma_S\kappa \leqslant 3.537 \times 10^{-33}$ | None |
| (c) Exponential $\rho_{Em}$ | (c1) Spatial separation $m = $ S | $\Gamma_S\kappa^2 \leqslant 1.25 \times 10^{-35}$ | $\Gamma_S\kappa^2 \leqslant 6.173 \times 10^{-32}$ | None |
| | (c2) Spacetime separation $m = $ ST | $\Gamma_S\kappa^2 \leqslant 2.812 \times 10^{-36}$ | $\Gamma_S\kappa^2 \leqslant 1.234 \times 10^{-32}$ | None |

See discussion in the main text on prior bounds.

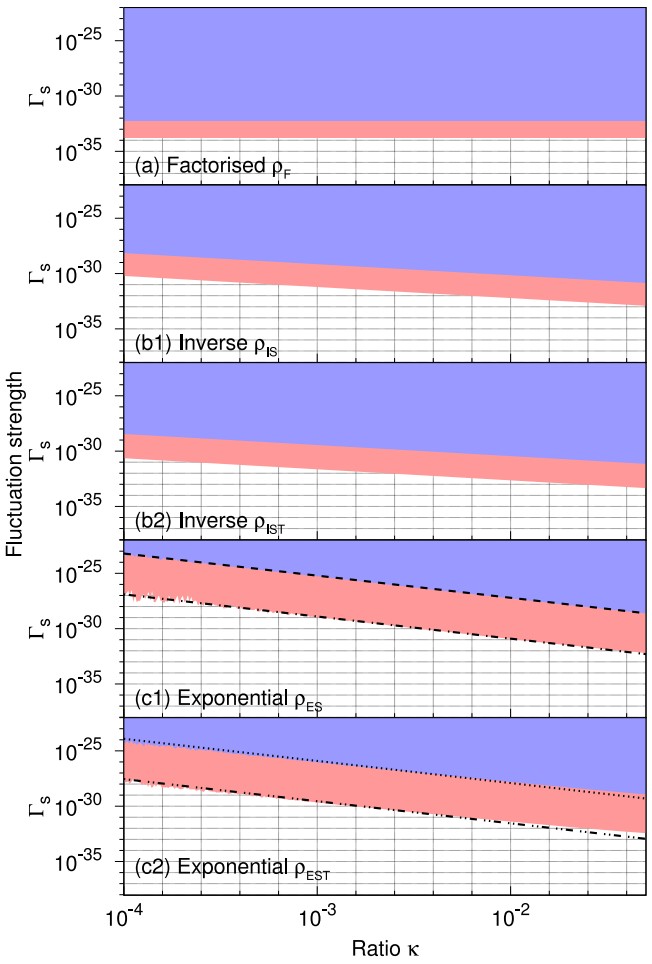

**Fig. 5 | Excluding parameter values using data from Holometer and QUEST.** Excluded region in the parameter space of $\Gamma_S$ and $\kappa = \ell_r/\mathcal{L}$ for correlation classes: (**a**) *factorised $\rho_F$*, (**b**) *inverse $\rho_{Im}$*, and (**c**) *exponential $\rho_{Em}$*, where $m = $ S ($m = $ ST) denotes correlations depending on spatial (spacetime) separation. Data from the Holometer excludes the regions shaded in blue and red; data from QUEST excludes the blue region only. The black dashed ($\Gamma_S\kappa^2 \leqslant 6.173 \times 10^{-32}$), dot-dashed ($\Gamma_S\kappa^2 \leqslant 1.25 \times 10^{-35}$), dotted ($\Gamma_S\kappa^2 \leqslant 1.234 \times 10^{-32}$), and dot-dot-dashed ($\Gamma_S\kappa^2 \leqslant 2.812 \times 10^{-36}$) lines denote curves fitted to the boundary of the excluded regions.

yielding the constraint $\alpha \leqslant 40$. The two orders of magnitude difference between the constraints on $\alpha$ obtained using data from the Holometer can be explained by a corresponding difference between the magnitude of the CSD and PSD from the same experiment. In order to use the CSD data, an implicit assumption that the correlation length is larger than the distance between the two interferometers is made.

We avoid such an assumption by restricting our comparison to the PSD data of the single interferometer. In subclass (b2), there are no constraints from prior work.

*(c) Exponential correlation function:* The constraints are obtained by setting $S(\nu = \nu_i) \leqslant S_{\text{meas}}(\nu = \nu_i)$ at every $\nu_i$. Here we numerically compute $S(\nu = \nu_i) = \Gamma_S\ell_r\mathcal{L}^2 S_C(\nu = \nu_i)/c$ using Eqs. (74-76) of the Supplementary Information. A numerical fit to such constraints yields the expressions listed in Table 1. In this class, there are no constraints from prior work.

## Discussion

We have developed a methodology to systematically compute interferometric output signal power spectral densities, produced by statistically defined spacetime fluctuations (SFs), under an explicit list of assumptions. Using this methodology, we have compiled the interferometric output signals due to SFs in three correlation classes, for Michelson laser interferometers with and without arm cavities. This allows us to identify characteristic signatures in spectral densities for the different classes of correlation functions of SFs. We also find that (1) the laboratory-scale QUEST and GQuEST will have the broad bandwidth needed to observe all the characteristic signatures, while (2) LIGO is better suited for detecting the bare presence or absence of SFs. By comparing with the experimental data from the Holometer and QUEST, we have also constrained the strength and correlation scale of the SFs in the classes (a)–(c), which tallies with constraints obtained in contemporary work[16].

Moreover, our methodology enables unambiguous computation of interferometric signals for other (current or future) theories of gravity just from the correlation function of the SFs and the geometry of the interferometer. It can also be applied to compute interferometric signals to search for stochastic gravitational waves[12,13] or dark matter[47]. Lastly, our methodology may be applied in instrumental 'noise hunting' or calibration efforts for interferometers, for cases where the noise or calibration signal can be described as metric or phase fluctuations[48–50] along the light path.

Future work could apply our methodology to other gravity models, such as continuous spontaneous localisation and Diósi-Penrose models, to constrain their free parameters using interferometric experiments. Further research could also extend our methodology to include more complicated interferometer configurations and multi-interferometer cross-correlations, which will then be used to further constrain the parameters of SFs.

## Methods
### Modelling light propagation
To investigate the effects of a fluctuating spacetime, we consider an isotropic spacetime metric $g^{\beta\gamma}$ ($\beta, \gamma = 0, 1, 2, 3$) of the form

$$g^{00} = -1 + 2w(\mathbf{r}), \quad g^{ij} = \delta_{ij}, \quad g^{0i} = g^{i0} = 0, \quad (4)$$

where $i, j = 1, 2, 3$ and $w(\boldsymbol{r})$ is a random process in $\boldsymbol{r} \equiv (t, x, y, z)$, with $w \ll 1$. Here 0 (resp., $i \in \{1, 2, 3\}$) corresponds to the timelike (resp., spacelike) component. While we consider this specific form in this work, our approach can encompass general fluctuations in every $g^{\beta\gamma}$ (see Sec. I of the Supplementary Information for details). To model the propagation of light of frequency $\Omega$ and wavelength $\lambda = 2\pi c/\Omega$ in the given spacetime manifold, we solve for the electromagnetic tensor in the relativistic wave equation (RWE) (see Sec. I of the Supplementary Information with ref. [51] therein for details). This is subject to the following assumptions.

**Assumption (i)** *Setting length and time scales:* The correlation scales in length and time of the metric fluctuations $w$ need to be longer than $\lambda$ and $2\pi/\Omega$ respectively. This effectively sets the wavelength as the smallest length scale in the system, i.e., the *eikonal approximation*. This allows us to neglect diffraction due to the fluctuations.

Applying the *eikonal approximation* ($k = 2\pi/\lambda \to \infty$), the RWE reduces to the light propagating along the null geodesic for any general spacetime metric[52]. For light propagating along, say, the $z$-axis, the electric field solution for the RWE (see Sec. I of the Supplementary Information for details) is

$$\vec{E}(\boldsymbol{r}(t)) = \vec{E}_{in}(x, y)e^{ik\Phi(\boldsymbol{r}(t))}, \qquad (5)$$

where

$$\Phi(\boldsymbol{r}(t)) = ct - z + \Phi_{\mathrm{F}}(\boldsymbol{r}(t)), \qquad (6a)$$

$$\Phi_{\mathrm{F}}(\boldsymbol{r}(t)) = c \int_0^t \mathrm{d}t' \, w(\boldsymbol{r}(t')), \qquad (6b)$$

and $\vec{E}_{in}(x, y)$ is the input transverse profile of the beam.

**Assumption (ii)** *Slowly varying envelope approximation (SVEA):* The above solution (5) also uses the SVEA, which is consistent with the eikonal approximation. It assumes a very small rate of metric-fluctuation-induced phase fluctuations $\partial_t \Phi_{\mathrm{F}} \ll c$ and $\partial_i \Phi_{\mathrm{F}} \ll 1$ ($i = x, y, z$).

We now introduce the assumptions on the random SF process $w$.

**Assumption (iii)** *Stationarity:* $w(\boldsymbol{r})$ is a stationary Gaussian random process with the expectation values,

$$\overline{w} = 0, \quad \text{and} \qquad (7)$$

$$\overline{w(t_1, \vec{r}_1)w(t_2, \vec{r}_2)} = \Gamma_{\mathrm{S}}\,\rho(ct_{12}, \vec{r}_{12}). \qquad (8)$$

Here $X_{12} = X_1 - X_2$ ($X = t, \vec{r}$) with $\vec{r}_i \equiv \{x_i, y_i, z_i\}$ ($i = 1, 2$) and $\Gamma_{\mathrm{S}}$ is the strength of the SFs. Both $\Gamma_{\mathrm{S}}$ and $\rho$ are dimensionless, consistent with $w$ being dimensionless. Subsequently, we consider the correlation function $\rho(ct_{12}, \vec{r}_{12})$ to be from the classes (a)–(c) to obtain the corresponding MLI output as illustrated in Fig. 1.

**Assumption (iv)** *Isotropy:* The two-point correlation function $\rho$ is isotropic in space, i.e.,

$$\begin{aligned} \rho\big(\delta_0, \{\delta_1, \delta_2, \delta_3\}\big) &= \rho\big(\delta_0, \{\delta_3, \delta_1, \delta_2\}\big) \\ &= \rho\big(\delta_0, \{\delta_3, \delta_2, \delta_1\}\big) = \ldots, \end{aligned} \qquad (9)$$

for any separation $\delta_\mu$ ($\mu = 0, 1, 2, 3$). We consider the correlation length of $\rho$ to be $\ell_r$ for all three spatial dimensions, and the correlation time to be $\ell_r/c$.

## Correlation classes

We now list the different correlation classes, which completes the description of the spacetime fluctuations.

(a) *Factorised correlation function:* This class of correlation functions of the form

$$\rho_{\mathrm{F}}\big(c\Delta_{\mathrm{t}}, \vec{\Delta}_{\mathrm{r}}\big) = -\left(\frac{\|\vec{\Delta}_{\mathrm{r}}\|}{\ell_r}\right)\left(\frac{\ell_r\,\delta(\Delta_{\mathrm{t}})}{c}\right), \qquad (10)$$

is motivated by the Oppenheim model[10], where $\vec{\Delta}_{\mathrm{r}}$ is any 3-vector in space and $\Delta_{\mathrm{t}}$ is any time interval. We note that multiplicative factors consistent with Assumption (iv) are suitably introduced such that the delta-correlated $\rho_{\mathrm{F}}$ remains dimensionless. Also, it is evident that $\rho_{\mathrm{F}}$ is independent of $\ell_r$. Thus, the only parameter in this correlation class is the fluctuation strength $\Gamma_{\mathrm{S}}$ that scales the correlation function (Eq. (8)).

(b) *Inverse correlation functions:* This class, a subset of the polynomial decay of the correlations, is motivated by models such as those of Karolyhazy[35], and Zurek[16], as well as effective field theories[17]. We consider two sub-classes where the correlations decay as a function of spatial separation and spacetime separation, respectively, with the specific form of the correlation functions defining the subclass listed below.

(b1) Spatial separation:

$$\rho_{\mathrm{IS}}\big(c\Delta_{\mathrm{t}}, \vec{\Delta}_{\mathrm{r}}\big) = \frac{\ell_r}{\|\vec{\Delta}_{\mathrm{r}}\|}\Theta(\|\vec{\Delta}_{\mathrm{r}}\| - c|\Delta_{\mathrm{t}}|), \qquad (11)$$

where the Heaviside theta function $\Theta(\|\vec{\Delta}_{\mathrm{r}}\| - c|\Delta_{\mathrm{t}}|)$ vanishes for all pairs of time-like separated spacetime points. Such correlation is found in models that assume the fluctuations satisfy the wave equation[16,17,35].

(b2) Spacetime separation:

$$\rho_{\mathrm{IST}}\big(c\Delta_{\mathrm{t}}, \vec{\Delta}_{\mathrm{r}}\big) = \frac{\ell_r\,\Theta(\|\vec{\Delta}_{\mathrm{r}}\| - c|\Delta_{\mathrm{t}}|)}{\sqrt{\|\vec{\Delta}_{\mathrm{r}}\|^2 - c^2\Delta_{\mathrm{t}}^2}}. \qquad (12)$$

This is a generalisation of (b1). Carrying forward the step function from (b1), the sign convention in the spacetime separation is chosen such that the correlation remains a real-valued function.

While we use two parameters to characterise strength, $\Gamma_{\mathrm{S}}$ and the scale $\ell_r$, in this class, the function depends only on the product $\Gamma_{\mathrm{S}}\ell_r$, rendering the parameters degenerate.

(c) *Exponential correlation functions:* This class of correlation functions covers models motivated by entanglement between holographic degrees of freedom[15] or a mesoscopic interpretation of gravity[11]. We again list the correlation functions for two sub-classes based on

(c1) Spatial separation:

$$\rho_{\mathrm{ES}}\big(c\Delta_{\mathrm{t}}, \vec{\Delta}_{\mathrm{r}}\big) = e^{-\frac{\|\vec{\Delta}_{\mathrm{r}}\|}{\ell_r}}\Theta(\|\vec{\Delta}_{\mathrm{r}}\| - c|\Delta_{\mathrm{t}}|). \qquad (13)$$

(c2) Spacetime separation:

$$\rho_{\mathrm{EST}}\big(c\Delta_{\mathrm{t}}, \vec{\Delta}_{\mathrm{r}}\big) = e^{-\frac{\sqrt{\|\vec{\Delta}_{\mathrm{r}}\|^2 - c^2\Delta_{\mathrm{t}}^2}}{\ell_r}}\Theta(\|\vec{\Delta}_{\mathrm{r}}\| - c|\Delta_{\mathrm{t}}|). \qquad (14)$$

Class (c) is a true two-parameter model, unlike classes (a) and (b). Classes (b) and (c) cannot be factorised into spatial and temporal functions.

## MLI output signal PSDs

In an MLI (Fig. 3 (a)), light propagates from a laser source at the input port A to a detector at the output port B via the two perpendicular arms, denoted by C and D. The 50/50 lossless beamsplitter is denoted by BS and is taken as the origin of the reference frame in our computation. The beamsplitter and the end mirrors are assumed to be

perfect due to the very small optical losses that are typical in these high-precision interferometers. We can effectively assume the detector to be at the origin, as the effect of any phase fluctuations suffered by the light after interference at the BS is negligible compared to that of phase fluctuations incurred in the arms in practice. The arm length of the MLI without fluctuations is $\mathcal{L}$ and light-round-trip time $\tau_0 = 1/f_{\text{LRT}}$ (we recall that $(f_{\text{LRT}})^{-1} = 2\mathcal{L}/c$). The PSD of the optical path difference between the two arms (see Sec. II of the Supplementary Information for details) at the detector is then written as a cosine transform from time separation $\Delta_\tau$ to frequency $f$.

$$S(f) = \frac{c^2 \Gamma_S}{2\pi} \int_0^\infty d\Delta_\tau \left[ \sigma(\Delta_\tau) - \xi(\Delta_\tau) \right] \cos 2\pi f \Delta_\tau, \quad (15)$$

where

$$
\begin{aligned}
\sigma(\Delta_\tau) &= \int_0^{\tau_0} dt_1 \int_0^{\tau_0} dt_2\, \rho(c(t_1 + \Delta_\tau - t_2), 0, 0, s(t_1) - s(t_2)) \\
&= \int_0^{\tau_0} dt_1 \int_0^{\tau_0} dt_2\, \rho(c(t_1 + \Delta_\tau - t_2), s(t_1) - s(t_2), 0, 0),
\end{aligned}
\quad (16a)
$$

$$\xi(\Delta_\tau) = \int_0^{\tau_0} dt_1 \int_0^{\tau_0} dt_2 \rho(c(t_1 + \Delta_\tau - t_2), s(t_1), 0, -s(t_2)). \quad (16b)$$

Also, $s(t) = c\,t$ if $t \leqslant \tau_0/2$ and $s(t) = 2\mathcal{L} - ct$ if $t > \tau_0/2$. Assumptions (i)–(iv) are used to obtain Eq. (15), with Eq. (16a) using Assumption (iv). We reiterate that the expressions obtained in this subsection are true irrespective of the form of the correlation function $\rho$ and are not limited to classes (a)–(c). $\sigma$ arises from correlations in the spacetime metric fluctuations within an arm of the interferometer, and $\xi$ corresponds to correlations of the metric fluctuations between the two arms. Here, we have also assumed that the width of the light beams are of the order of the wavelength and negligible (see Sec. II of the Supplementary Information for details).

A response-function-based approach allows straightforward extension of our methodology to different MLI geometries and detection schemes. These include for instance, the cross spectral density (CSD) of the output signal from two co-located, co-aligned MLIs (see Sec. II and III of the Supplementary Information for details) and the signal PSD from an MLI with Fabry-Pérot arm cavities (see Sec. IV of the Supplementary Information for details) in the presence of SFs. Here it is important to note that both our approaches of obtaining the spectral densities, namely, based on correlation integral in Eq. (15) and the response function in Eq. (17) assume weak stationarity of the optical path difference. However, the existence of such a stationarity is not guaranteed, even when assuming the underlying SFs to be stationary and Gaussian. Therefore, when computing the PSDs, we check if the covariance of the optical path difference is positive-definite, to ensure weak stationarity of the optical path difference in our investigation. We find in class (c), this holds only when $\kappa = \ell_r/\mathcal{L} \ll 1$.

We thus rewrite the PSD in terms of the corresponding *interferometer response function* $\widetilde{\chi}_1(f, \vec{k}_1)$ as an integral over a 3-dimensional reciprocal space (see Sec. III of the Supplementary Information for details).

$$S(f) = \int d^3\vec{k}_1\, \Gamma_S\, \widetilde{\rho}(2\pi f, \vec{k}_1)\, \widetilde{\chi}_1(f, \vec{k}_1), \quad (17)$$

where $\widetilde{\rho}(2\pi f, \vec{k}_1)$ is given by

$$\widetilde{\rho}(2\pi f, \vec{k}_1) = \frac{1}{(2\pi)^4} \int d^3\vec{r}_{12} \int_{-\infty}^\infty dt_{12}\, \rho(ct_{12}, \vec{r}_{12})\, e^{-i\left(2\pi f t_{12} + \vec{k}_1 \cdot \vec{r}_{12}\right)}, \quad (18)$$

and the response function $\widetilde{\chi}_1(f, \vec{k}_1)$ of the MLI (Fig. 3 (a)) is

$$\widetilde{\chi}_1(f, \vec{k}_1) = \left(\frac{\mathcal{L}}{2}\right)^2 \left| C_x(f, \vec{k}_1) - C_z(f, \vec{k}_1) \right|^2, \quad (19)$$

with

$$C_j(f, \vec{k}_1) = e^{if T_+^{(j)}} \left\{ \text{Sinc}\left( f T_+^{(j)} \right) + e^{\frac{2\pi i f \mathcal{L}}{c}} \text{Sinc}\left( f T_-^{(j)} \right) \right\}, \quad (20)$$

$$T_\pm^{(j)}(f, \vec{k}_1) = \frac{\pi \mathcal{L}}{c} \left( 1 \pm \frac{c}{2\pi f} \vec{k}_1 \cdot \hat{e}_j \right), \quad (j = x, z). \quad (21)$$

It is interesting to contrast the general 4-dimensional Fourier transform considered in Eq. (18) with the plane wave expansions of *'metric perturbations'* considered in contemporary investigations of stochastic gravitational wave backgrounds[53]. These investigations consider stochastic gravitational waves as perturbations of the metric in the transverse-traceless gauge, and they require such perturbations to be a sum of plane waves that satisfy the wave equation. In contrast, we do not expect the stationary Gaussian SFs in the non-relativistic limit to satisfy the wave equation.

We also find the response function corresponding to an MLI with Fabry-Pérot arm cavities (see Fig. 3 (b) for the interferometer geometry; also see Sec. IV of the Supplementary Information for details). The signal PSD in terms of this response function is given by

$$S(f) = \widetilde{\chi}_{\text{FP}}(f) \int d^3\vec{k}_1\, \Gamma_S\, \widetilde{\rho}(2\pi f, \vec{k}_1)\, \widetilde{\chi}_1(f, \vec{k}_1), \quad (22)$$

where the Fabry-Pérot cavity response is given by

$$\widetilde{\chi}_{\text{FP}}(f) = T_M^4 \left(\frac{1}{1 - \sqrt{R_M}}\right)^4 \left(\frac{1}{1 + R_M - 2\sqrt{R_M} \cos(2\pi f / f_{\text{LRT}})}\right), \quad (23)$$

and $T_M = 1 - R_M$ is the power transmissivity, and $R_M$ is the power reflectivity of the mirror M rendering the arm cavity in either arm.

To assess and compare the behaviour of the computed signal PSDs for the different correlation function classes, we consider the dimensionless frequency and PSDs.

$$\nu \doteq \pi f / (2 f_{\text{LRT}}) \quad (24)$$

$$S_{\text{NC}}(\nu) \doteq \left(\frac{c}{\Gamma_S \mathcal{L}^3}\right) S(f), \text{ and} \quad (25)$$

$$S_C(\nu) \doteq \left(\frac{c}{\Gamma_S \ell_r \mathcal{L}^2}\right) S(f). \quad (26)$$

Note that we have two types of dimensionless PSDs: $S_{\text{NC}}$ independent of $\ell_r$, catering to correlation class (a), which is also independent of $\ell_r$, and $S_C$, which depends on $\ell_r$ for correlation classes (b) and (c). From Eqs. (15, 17 and 22), it is clear that both $S_{\text{NC}}(\nu)$ and $S_C(\nu)$ are independent of $\Gamma_S$.

## Data availability

The datasets from QUEST and the Holometer were analysed during the current study. The Holometer dataset belongs to Fermilab and is shared publicly as mentioned in https://holometer.fnal.gov/. A limited access to the QUEST dataset was provided to the authors by Prof Hartmut Grote from the University of Cardiff, for the sole purpose of analysis. Therefore, this dataset cannot be shared publicly by the authors of this manuscript.

## Code availability

We do not use custom code or mathematical algorithms that are central to the conclusions of this work.

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

## Acknowledgements

We thank Hartmut Grote, Jonathan Oppenheim and Ohkyung Kwon for extensive discussions and suggestions crucial to this work. We also thank Vincent Lee for clarifications on the Pixellon model. We thank the QUEST team for sharing data for comparison with our results. BS thanks Dr. V. Balakrishnan for vital clarifications and discussions on aspects of stationarity. BS and AD acknowledge the UK STFC "Quantum Technologies for Fundamental Physics" programme (Grant Numbers ST/T006404/1, ST/W006308/1 and ST/Y004493/1) for support. BS also acknowledges the support of the Leverhulme Trust under research grant ECF-2024-124. SMV acknowledges the support of the Leverhulme Trust under research grant RPG-2019-022.

## Author contributions

B.S. contributed to methodology, formal analysis, data curation, visualisation, writing—original draft and editing. S.M.V. contributed to conceptualisation, methodology, formal analysis, writing—review and editing. A.D. contributed to conceptualisation, methodology, formal analysis, writing—review and editing, supervision and project administration.

## Competing interests

The authors declare no competing interests.
