## [Transparent Peer Review file · Nature Communications]

Signatures of Correlation of Spacetime Fluctuations in Laser Interferometers

Corresponding Author: Dr Sharmila Balamurugan

Version 0:

Reviewer comments:

Reviewer #1

(Remarks to the Author)

In their manuscript "Signatures of Correlation of Space Time Fluctuations in Laser Interferometers" the authors discuss the signatures of spacetime fluctuations in interferometer experiments and in particular investigate the characteristics of the power spectral density associated with different type of correlations in the fluctuations in the geometry of space and time. They present a careful analysis of three different type of correlations: factorized, inverse, and exponential correlations, where in the last two cases they also distinguish between spatial and spacetime separations. The results are written in terms of an output signal that also takes into account the response function associated to the interferometer. The analytical and numerical methods are sound, reasonably standard and are explained in a clear fashion. The obtained output signals for the power spectral density are presented concisely in three plots that clearly show the qualitative and quantitative differences between the different models.

The significance of this work lies in understanding how to analyse the data that will be obtained in present day and future laser interferometer experiments. One of these experiments is the QQUEST experiment, which is specifically designed for the purpose of detecting spacetime fluctuations. The possibility of detecting such signatures is of great scientific interest, and even when no positive signatures are found, the findings will provide important information about the theoretical models that are being developed. I find the paper well written and the results of high enough relevance and interest to recommend publication in Nature Communications.

(Remarks on code availability)

Reviewer #2

(Remarks to the Author)

The topic investigated in this manuscript is very significant: quantum-gravity-motivated studies of spacetime fluctuations have a very long tradition, with a more intense effort over the last 25 years, ignited mainly by Nature 398 (1999) 216

This is a research program facing challenges of two types: on the experimental side the challenge originates from the fact that the relevant effects are expected to be very small while on the theory side one faces the challenge that the structure of the quantum-gravity problem, while pointing rather clearly to the presence of spacetime fluctuations, provides little guidance on how spacetime fluctuations should be formalized and modeled (therefore many alternative formalizations are being studied).

I formed a positive opinion of this manuscript since its results and observations will prove valuable for the phenomenology of some of the attempted formalizations of spacetime fluctuations.

I recommend the manuscript for publication.

(Remarks on code availability)

Reviewer #3

(Remarks to the Author)

The manuscript "Signatures of Correlation of Spacetime Fluctuations in Laser Interferometers" by Sharmila et al. analyzes the effect of spacetime fluctuations on laser interferometers LIGO, QUEST and GQuEST. In particular, they consider three possible classes for the correlations of the spacetime fluctuations, that encompass different models (collapse, semiclassical gravity, decoherence, effective field theory, holographic dof, etc). They find the corresponding PSDs (power spectral densities), which are expressed in terms of the strength (Γ) and characteristic length (l_r) of the correlations, the interferometer arm length (L) and the scaled frequency (ν). Depending on the specific class (and subclass) of correlations, the PSDs provide different qualitative behaviors, which would allow to identify the origin of spacetime fluctuations.

Unfortunately, it is not clear how much the proposed analysis can be experimentally verified. Although citing and constructing ad-hoc PSD for the three experiments mentioned above, the authors completely neglect the direct comparison of their results with experimental data. Since the parameters Γ and l_r are free, I suppose such a comparison would have led to experimental bounds on their possible values. I believe this is a strong shortcoming of the manuscript.

Minor comments:

- Introduction: there are several semiclassical models of gravity that are precedent than the cited [5,6] that are worth mentioning here. For example, Kafri-Taylor-Milburn, Tilloy-Diosi, Schrodinger-Newton etc
- Introduction, in place of [2] the authors might want to cite the original papers by Diosi and by Penrose.
- Introduction, when introducing the classes of correlations (pg 2, top of the left column), the general description does not cover the subclasses b1 and c1.
- Eq.(6) and below, the authors might want to use the standard notation for the indices: 0 for temporal component, 1-3 for the spatial ones.
- Eq.(8), I suppose Θ is a Heaviside θ function. Please define
- Above Eq.(12), what is physically τ_0 ?
- Eq.(20), T_M and R_M are not defined.
- Fig.2, are the wrinkles around $\nu=1$ of any interest?
- Why is the class (a) not discussed in the section "Detecting SFs"?

I will be happy to review the manuscript again, after the authors revise it.

(Remarks on code availability)

Version 1:

Reviewer comments:

Reviewer #3

(Remarks to the Author)

I am happy with the changes made and I can recommend the manuscript for publication.

Point-wise response to Reviewers

We thank Reviewers 1 and 2 for their recognition of our work and their recommendation to publish. Our pointwise response to the comments of reviewer 3 is as follows.

REVIEWER 3:

Major comment 1: “Unfortunately, it is not clear how much the proposed analysis can be experimentally verified. Although citing and constructing ad-hoc PSD for the three experiments mentioned above, the authors completely neglect the direct comparison of their results with experimental data. Since the parameters Γ_s and ℓ_r are free, I suppose such a comparison would have led to experimental bounds on their possible values. I believe this is a strong shortcoming of the manuscript.”

Response:

We agree with the referee that the suggested comparison adds significant scientific value to the current manuscript.

We have therefore carried out a comparison of our expected interferometric output signal PSD with the data from the Holometer [1, Fig. 12] and QUEST [2, Fig. 4]. We do not yet consider data from LIGO or other gravitational-wave (GW) detectors, as these instruments operate and record data in a bandwidth that spans a relatively small range of ν and does not include the highest magnitude of the signal PSD, which is at $\nu = \pi/2$. The comparison to data from QUEST and the Holometer has been reported in a new subsection titled “Comparison with experimental data”. We have also added both Fig. 5 plotting the exclusion plots in the parameter space and Table 1 listing the obtained constraints on Γ_s and κ for the different correlation classes. We have also compared our constraints with pre-existing ones.

Related change in the manuscript:

A new subsection titled “Comparison with experimental data”, Fig. 5 and Table 1 have been added to the main manuscript.

Minor comment 1: “Introduction: there are several semiclassical models of gravity that are precedent than the cited [5,6] that are worth mentioning here. For example, Kafri-Taylor-Milburn, Tilloy-Diosi, Schrodinger-Newton etc”

Response:

References [5-9] in the revised manuscript have been added. They are as listed below.

- [5] L. Rosenfeld, On quantization of fields, *Nucl. Phys.* **40**, 353 (1963).
 - [6] R. Ruffini and S. Bonazzola, Systems of self-gravitating particles in general relativity and the concept of an equation of state, *Phys. Rev.* **187**, 1767 (1969).
 - [7] J. R. van Meter, Schrödinger–newton ‘collapse’ of the wavefunction, *Classical Quant. Grav.* **28**, 215013 (2011).
 - [8] D. Kafri, J. M. Taylor, and G. J. Milburn, A classical channel model for gravitational decoherence, *New J. Phys.* **16**, 065020 (2014).
 - [9] A. Tilloy and L. Diósi, Sourcing semiclassical gravity from spontaneously localized quantum matter, *Phys. Rev. D* **93**, 024026 (2016).
-

Minor comment 2: “Introduction, in place of [2] the authors might want to cite the original papers by Diosi and by Penrose.”

Response:

References [39-41] in the revised manuscript have been added. They are as listed below.

- [39] L. Diósi, Gravitation and quantum-mechanical localization of macro-objects, *Physics Letters A* **105**, 199 (1984).
- [40] L. Diósi, Models for universal reduction of macroscopic quantum fluctuations, *Phys. Rev. A* **40**, 1165 (1989).

[41] R. Penrose, On gravity's role in quantum state reduction, *Gen. Relat. Gravit.* **28**, 581 (1996).

Minor comment 3: “Introduction, when introducing the classes of correlations (pg 2, top of the left column), the general description does not cover the subclasses b1 and c1.”

Response:

We have added to and corrected the statement in the Introduction. The clause “, with subclasses refining the definition of the separation as either a spatial or a spacetime separation.” has been added to the definition of class (b). The clause “, with subclasses similar to those in (b).” is added to the definition of class (c).

Minor comment 4: “Eq.(6) and below, the authors might want to use the standard notation for the indices: 0 for temporal component, 1-3 for the spatial ones”

Response:

Equation (6) has been modified accordingly.

Minor comment 5: “Eq.(8), I suppose Theta is a Heaviside theta function. Please define”

Response:

Yes. This comment has been addressed by adding “where the Heaviside theta function $\Theta(\|\vec{\Delta}_r\| - c|\Delta_t|)$ vanishes for all pairs of time-like separated spacetime points.” below Eq. (8).

Minor comment 6: “Above Eq.(12), what is physically τ_0 ?”

Response:

τ_0 denotes the light-round-trip time. This has been added above Eq. (12).

Minor comment 7: “Eq.(20), T_M and R_M are not defined.”

Response:

We have now defined the coefficients by adding “and $T_M = 1 - R_M$ is the power transmissivity and R_M is the power reflectivity of the mirror M rendering the arm cavity in either arm.” below Eq. (20). We have also added a definition of the power transmissivity to the figure caption of Fig. 1 to improve clarity.

Minor comment 8: “Fig.2, are the wrinkles around $\nu = 1$ of any interest?”

Response:

These wrinkles are due to the interferometer response and is unrelated to the specific nature of the spacetime fluctuations. We have added a footnote “Another feature that is present in the scaled PSDs of the three classes, is the oscillatory behaviour at $\nu > 1$. This feature originates from the interferometer response [3, Eqs. (49)-(50)] and is not a characteristic signature of the different classes.” in the paragraph beginning “As shown above, ...” in Pg. 6.

Minor comment 9: “Why is the class (a) not discussed in the section “Detecting SFs”?”

Response:

In both classes (a) and (b), the signal strength in LIGO is enhanced by Fabry-Pérot cavity response. As the reasons behind are identical, the following line has been added to item 2 in the “Detecting SFs” subsection. “A

similar argument holds for class (a) as its S_{NC} for an MLI without arm cavities is also independent of \mathcal{L} .”

References

- [1] A. Chou, H. Glass, H. R. Gustafson, C. Hogan, B. L. Kamai, O. Kwon, R. Lanza, L. McCuller, S. S. Meyer, J. Richardson, C. Stoughton, R. Tomlin, and R. Weiss, “The holometer: an instrument to probe planckian quantum geometry,” *Classical Quant. Grav.*, vol. 34, p. 065005, feb 2017.
- [2] A. Patra, L. Aiello, A. Ejlli, W. L. Griffiths, A. L. James, N. Kuntimaddi, O. Kwon, E. Schwartz, H. Vahlbruch, S. M. Vermeulen, K. Kokeyama, K. L. Dooley, and H. Grote, “Broadband limits on stochastic length fluctuations from a pair of table-top interferometers,” *Phys. Rev. Lett.*, vol. 135, p. 101402, Sep 2025.
- [3] See Supplemental Material for details regarding the field equations in a fluctuating spacetime, deriving the spectral densities and the corresponding response functions for the two types of setups (aLIGO and Holometer), and the behaviour of the PSD for the different correlation function classes.